# Facilitators and barriers impacting in-hospital Trauma Quality Improvement Program (TQIP) implementation across country income levels: a scoping review

George Kapanadze [1], Johanna Berg [1,2] Yue Sun,[1] Martin Gerdin Wärnberg [1,3]

¹Department of Global Public Health, Karolinska Institute, Stockholm, Sweden
²Emergency and Internal Medicine, Skånes universitetssjukhus Malmö, Malmo, Sweden
³Function Perioperative Medicine and Intensive Care, Karolinska University Hospital, Stockholm, Sweden

**Correspondence to**
Dr George Kapanadze;
george.kapanadze@stud.ki.se

## ABSTRACT

**Objective** Trauma is a leading cause of mortality and morbidity globally, disproportionately affecting low/middle-income countries (LMICs). Understanding the factors determining implementation success for in-hospital Trauma Quality Improvement Programs (TQIPs) is critical to reducing the global trauma burden. We synthesised topical literature to identify key facilitators and barriers to in-hospital TQIP implementation across country income levels.

**Design** Scoping review.

**Data sources** PubMed, Web of Science and Global Index Medicus databases were searched from June 2009 to January 2022.

**Eligibility criteria** Published literature involving any study design, written in English and evaluating any implemented in-hospital quality improvement programme in trauma populations worldwide. Literature that was non-English, unpublished and involved non-hospital TQIPs was excluded.

**Data extraction and synthesis** Two reviewers completed a three-stage screening process using Covidence, with any discrepancies resolved through a third reviewer. Content analysis using the Consolidated Framework for Implementation Research identified facilitator and barrier themes for in-hospital TQIP implementation.

**Results** Twenty-eight studies met the eligibility criteria from 3923 studies identified. The most discussed in-hospital TQIPs in included literature were trauma registries. Facilitators and barriers were similar across all country income levels. The main facilitator themes identified were the prioritisation of staff education and training, strengthening stakeholder dialogue and providing standardised best-practice guidelines. The key barrier theme identified in LMICs was poor data quality, while high-income countries (HICs) had reduced communication across professional hierarchies.

**Conclusions** Stakeholder prioritisation of in-hospital TQIPs, along with increased knowledge and consensus of trauma care best practices, are essential efforts to reduce the global trauma burden. The primary focus of future studies on in-hospital TQIPs in LMICs should target improving registry data quality, while interventions in HICs should target strengthening communication channels between healthcare professionals.

## STRENGTHS AND LIMITATIONS OF THIS STUDY

⇒ The present scoping review was based on a comprehensive search of published literature that included a robust screening process involving multiple peer reviewers to protect against bias.

⇒ We used the Consolidated Framework for Implementation Research to ground collected evidence in an established theory.

⇒ The generalisability of our results was impacted by grouping literature based on country income level, as we did not investigate potential differences in available resources within countries (eg, regions and healthcare institutions).

⇒ The lack of quality assessment involved in scoping review design limits the implications of gathered results, given the focus on mapping available literature and identifying knowledge gaps.

## INTRODUCTION

Trauma is the clinical entity composed of physical injury and the body's associated response.[1] Trauma is a leading cause of death and disability worldwide.[2] To improve in-hospital care, multiple Trauma Quality Improvement Programs (TQIPs) have been developed, typically originating in high-income countries (HICs). Key in-hospital TQIPs highlighted by the WHO in their 2009 guidelines included techniques such as trauma registries along with mortality and morbidity conferences.[3] The implementation of TQIPs has been attributed to significant improvements in care quality and outcomes in HICs.[4]

Quality improvement is designed to achieve positive change in a specific process,[5] which can be highly dependent on the perspectives, attitudes, and behaviours of patients and healthcare professionals in their local context.[6] Although quality improvement aims to improve the quality of care, organisations often struggle with its implementation.[7]

While implementation research has improved significantly, there remain substantial gaps in knowledge and understanding of factors that facilitate or impede efforts to implement quality improvement in healthcare organisations.[8]

Despite research on developing in-hospital TQIPs steadily rising, little context-relevant guidance exists to help policymakers set priorities for implementation.[9] A recent study found a lack of in-depth examination of barriers and facilitators to TQIP implementation in low/middle-income countries (LMICs).[10] This exemplifies the need for further research to assess trauma systems in LMICs to inform health system strengthening for trauma care.[4] Thus, this scoping review aimed to identify key facilitators and barriers to in-hospital TQIP implementation across country income levels.

## METHODS
### Study design
We conducted a scoping review to map the available evidence on factors influencing in-hospital TQIP implementation. The scoping review is reported in accordance with the Preferred Reporting Items for Systematic Reviews and Meta-Analyses extension for Scoping Reviews.[11] A scoping review was most appropriate due to the broad nature of the topic along with the wide variety of potential factors impacting quality improvement implementation in differing income settings. We defined country income level as the income level of each included country based on World Bank classification at the time of publication.[12]

### Search strategy
The search strategy was developed with a librarian using keywords, synonyms and medical subject headings as shown in online supplemental file 1. The review searched three databases, *PubMed*, *Web of Science* and *Global Index Medicus*, for studies from June 2009 to January 2022. The search was limited to publications from June 2009 onwards in line with the WHO release on Guidelines for TQIPs.[3]

### Study selection
Given the review's broad scope, all in-hospital TQIPs were considered, along with studies of any design. The screening process was completed by two authors, assisted by a third reviewer. The following inclusion criteria were used: any English-language studies evaluating facilitators and barriers to implementation of system-wide in-hospital quality improvement programmes in trauma populations worldwide. Studies were excluded if they were: non-English, unpublished, if they described projects where implementation was not a focus or if they discussed non-hospital TQIPs.

### Data charting
Data extraction was performed using a predetermined form addressing study-identifying information:

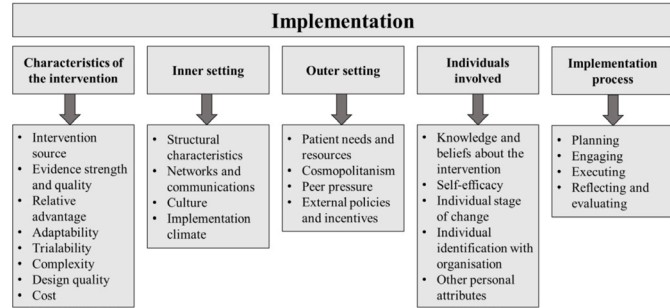

**Figure 1** Consolidated Framework for Implementation Research conceptual framework outline (adapted from Damschroder *et al*).[13]

geographical location; population and study setting; country income level at the time of publication; data collection methods; quality improvement intervention type and duration; implementation facilitators and barriers; implementation outcome and study funding sources. Appropriate quotations were identified and extracted through deductive content analysis. We did not evaluate the evidence underlying the identified TQIPs, as our focus was on facilitators and barriers to implementing a TQIP once attempted.

### Summarising and reporting results
Facilitator and barrier selection was guided by the Consolidated Framework for Implementation Research (CFIR) (illustrated in figure 1). The framework is composed of five domains: intervention characteristics, outer setting, inner setting, individual characteristics and implementation process.[13] These domains provide a practical guide for systematically assessing potential facilitators and barriers to TQIP implementation[14] while offering insights to explain differences across income levels.[15]

### Patient and public involvement
None.

## RESULTS
The literature search yielded 3923 studies, 3291 studies passed title and abstract screening, while 170 reports were identified for full-text screening. The number of extracted studies was 28,[10 16–42] with details on study selection found in figure 2. Most of the studies originated in upper middle-income countries (7; 25%), followed by low-income (5; 18%), high-income (4; 14%) and lower middle-income (4; 14%) countries. The remaining studies (8; 29%) included multicountry analyses that spanned across income levels. The most represented continents in the review were Africa and Asia, with South Africa (11; 39%) and Pakistan (9; 32%) appearing most frequently. Most studies solely discussed trauma registries as the prominent in-hospital TQIP (18; 64%), while a small number evaluated two or more TQIPs in their reports (5; 18%). Full study characteristic details are found in online supplemental file 2.

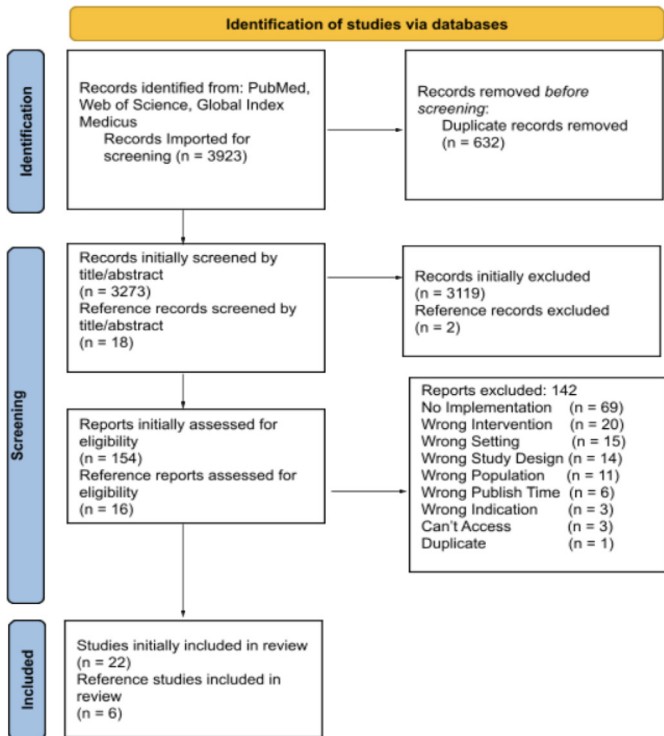

**Figure 2** Study selection.

The CFIR domains identified most in the literature were intervention characteristics and inner setting. A summary of facilitators and barriers according to the CFIR is shown in tables 1 and 2.

### Domain 1: characteristics of the intervention

Open dialogue with key stakeholders facilitates the implementation process.[10 16 17 33] A multicentre digital trauma registry implementation in Malawi revealed that consistent communication between Ministry of Health stakeholders and the research team contributed to the success of the registry.[16] Competing interests between groups acted as a barrier to establishing communication channels between stakeholders.[10 17 21] The absence of a vision for trauma care was highlighted by a custodian from a multihospital in a developing country as there was a need to 'show the vision of better trauma care to lots of people, because trauma is a very low priority for most clinicians, and it happens to poor people'[21] (O'Reilly et al, p119).

Another key theme was the importance of adaptability. A focus on tailoring the intervention to the local context to ensure success and sustainability was apparent.[10 17–19 25 27 34 35] The implementation of a multihospital injury surveillance programme in Mozambique was driven by local data collectors that enhanced the sustainability of the project.[18] Tailoring programmes to local conditions is often made difficult by individuals' resistance to change.[10 31] An educational self-management intervention in a trauma centre in the UK revealed that the roll-out of support was hindered by team members preferring to continue with established practices, which limited widespread adoption.[24]

Attention to simplicity is an implementation facilitator.[17 20 21 36] A trauma registry development study in LMICs emphasised the significance of keeping design simple through the standardisation of variables to enhance accuracy and understanding.[17] Standardisation requires widespread staff and stakeholder support, which is often lacking due to a limited desire for quality improvement. A survey of trauma care providers in the Andean region revealed that less than half of respondents 'ever witnessed a change occur in their institution', which influenced the 'current lack of motivation for quality improvement programs'[27] (LaGrone et al, p1992).

### Domain 2: outer setting

Prioritising patient care during implementation was widely addressed.[22–24 35] A Dutch evaluation of the implementation of the Transmural Trauma Care Model found that patients' experience improved as more focus was placed on day-to-day issues patients encountered.[22] However, time constraints were a key barrier to providing improved patient care.[10 16 22 24 31 32 34 37] A Peruvian study on the status of in-hospital TQIPs revealed the 'prevalence of dual practice'[10] (LaGrone et al, p966), surgeons working at both public and private institutions, meant that clinicians did not have time to effectively participate in emerging quality improvement initiatives.

Collaboration with national and international bodies was a facilitator to successful implementation.[17 23] A questionnaire with trauma registry stewards in LMICs showed a need to 'develop skills at political lobbying to induce support from the administration' to provide a platform for long-term aid[17] (Rosenkrantz et al, p2221). The lack of common guidelines was highlighted as a barrier.[10 17 21 28 32 38] A study exploring staff trauma experiences in LMICs underlined that organisational diversity across the region led to limited uniform policies which reduced communication among trauma facilities.[10]

### Domain 3: inner setting

Increased staff education and training strengthened the implementation climate of quality improvement interventions.[18–21 24 25 29 35 36 39 41] An integrated self-management support paper emphasised successful training as accessible to staff at all levels, with a focus on flexibility.[24] Professional hierarchies acted as a barrier to the continued education of all staff.[10 19 24 25] A training programme clinician survey found that traditional decision-making hierarchies acted as a major barrier to effective education as team members were 'reluctant to speak up to highlight a problem, clarify information, or question a senior's decision when they were concerned'[25] (Murphy p1151).

Implementation readiness was positively impacted by prioritising leadership buy-in.[10 17 20 35 39] An electronic registry in Pakistan highlighted obtaining senior management buy-in as a catalyst to improving trauma care quality.[39] Lack of stakeholder buy-in acted as a barrier to achieving progress in trauma care.[17 21] A trauma registry custodian from an LMIC hospital stated that a common

**Table 1** Summary of in-hospital TQIP implementation facilitators using the CFIR

| CFIR domain | CFIR construct | Facilitators | Quotes |
|---|---|---|---|
| Intervention characteristics | Relative advantage | Open dialogue with key stakeholders | 1. *Active cooperation and extensive input from Ministry of Health stakeholders*[16]<br>2. *Appealing to stakeholders higher up in the Department of Health*[17] |
| | Adaptability | Tailoring intervention to local context | 1. *All data collection was carried out by local Mozambican health workers*[18]<br>2. *Participants highlighted that patient care should be adapted to local circumstances*[19] |
| | Complexity | Focus on simplicity | 1. *Balance between sufficient detail and simplicity of the data collection process*[20]<br>2. *Less is more when it comes to trauma registry*[21] |
| Outer setting | Patient needs and resources | Prioritising patient care | 1. *More focus on the everyday things that patients have to deal with*[22]<br>2. *Harnessing…clinical enthusiasm to improve trauma care*[21] |
| | External policies and incentives | Collaboration with national/international bodies | 1. *Increased focus on trauma academics through regional and international collaborations*[23]<br>2. *Seek potential collaborators with interest*[17] |
| Inner setting | Implementation climate | | |
| | Learning climate | Staff education and training | 1. *Engage nursing and medical staff in multidisciplinary training*[24]<br>2. *Training needs to include strategies that address fundamental interpersonal communication*[25] |
| | Readiness for implementation | | |
| | Leadership engagement | Prioritise buy-in | 1. *Leadership was identified as having a clear impact on team factors*[25]<br>2. *The injury prevention coordinators' work was influenced by the support from direct managers and senior leadership*[26] |
| | Available resources | Diversify resource sources | 1. *External training resources, such as national or state conferences, were important to allow injury prevention champions to connect*[26]<br>2. *Broadening the scope of the registry to apply for additional grants*[17] |
| Characteristics of individuals | Knowledge and beliefs about the intervention | Generate understanding of quality improvement | 1. *There is a need for increased dissemination of QI programs as a means of empowering local providers to participate in their health systems*[27]<br>2. *Formulate very clearly a vision, mission and strategy that everyone understands*[21] |
| | Other personal attributes | Communication training | 1. *Important to find ways to sustain changes in clinician communication style*[24]<br>2. *Training needs to include strategies that address fundamental interpersonal communication alongside other solutions*[25] |
| Process of implementation | Reflecting and evaluating | Strengthening data quality | 1. *Increasing real and perceived efficacy of mortality & morbidity conferences…through the establishment of standardised case selection criteria*[27]<br>2. *Regular audits and staged validation of data*[21] |
| | Engaging | | |
| | Champions | Identifying dedicated staff | 1. *Sometimes I just get inspired by something that occurs to me as nothing is being done about it currently*[26]<br>2. *Identify a person with passion and zeal to drive the effort*[17] |

CFIR, Consolidated Framework for Implementation Research; TQIP, Trauma Quality Improvement Program.

opinion was that 'even basic care is not being given to the patients…how are we going to sort of spend for a registry?'[21] (O'Reilly *et al*, p122).

Diversifying resource sources is fundamental for successful implementation.[17 29 42] Respondents in a resource-constrained setting highlighted demand should

**Table 2** Summary of in-hospital TQIP implementation barriers using the CFIR

| CFIR domain | CFIR construct | Barriers | Quotes |
|---|---|---|---|
| Intervention characteristics | Relative advantage | Limited desire for quality improvement initiatives | 1. *A "so what" attitude amongst stakeholders from other nearby facilities*[17]<br>2. *Current lack of motivation for quality improvement programs in Latin America*[27] |
| | Adaptability | Resistance to change | 1. *Surgeon's preference for autonomy and self-reliance over standardisation: "I do it my way, you do it yours"*[10]<br>2. *Allied health professionals identified the barriers of…'the way things are always done'*[24] |
| | Complexity | Competing interests between groups | 1. *Institutional and national prioritisation of other patient care objectives*[10]<br>2. *Trauma is a very low priority for most clinicians*[21] |
| Outer setting | Patient needs and resources | Staff time constraints | 1. *Staff have multiple demands on their time*[16]<br>2. *Surgeon spends less time with a patient, he can proceed with the next patient*[22] |
| | External policies and incentives | Lack of common guidelines | 1. *Absence of uniform policies, guidelines, and protocols*[19]<br>2. *No standards or best practices in place for the collection of data*[28] |
| Inner setting | Implementation climate | | |
| | Learning climate | Professional hierarchies | 1. *Senior staff attempting to maintain the status quo in order to fly under the radar*[10]<br>2. *Limited ability to engage leaders*[19] |
| | Readiness for implementation | | |
| | Leadership engagement | Lack of buy-in | 1. *Some stakeholders…created barriers by prioritising other clinical issues*[17]<br>2. *Rest of the stakeholders and the doctors did not buy-in to this concept of using a registry*[21] |
| | Available resources | Limited resources | 1. *Hospitals cannot afford to hire specialised personnel solely to collect data*[29]<br>2. *Make do with the resources that you have, because the funding will not be there*[21] |
| Characteristics of individuals | Knowledge and beliefs about the intervention | Lack of staff belief in quality improvement interventions | 1. *Some initial scepticism from counterparts*[16]<br>2. *It's not fitting into anybody's career plans at this point in time…trauma happens to poor people*[30] |
| | Other personal attributes | Lack of staff knowledge | 1. *Almost half of newly-graduated physicians do not have experience working with this QI tool during their training*[31]<br>2. *Not a healthy understanding of how we do our work or what's needed to do our work*[26] |
| Process of implementation | Reflecting and evaluating | Poor data quality | 1. *Challenges reported included the high rates of missing data*[18]<br>2. *A major hurdle is data quality of clinical records*[32] |
| | Engaging | | |
| | Champions | Overworked staff | 1. *Human resources to meet volume of data collection were underestimated*[19]<br>2. *Constant suggesting by the faculty to use overworked residents to collect trauma data*[17] |

CFIR, Consolidated Framework for Implementation Research; TQIP, Trauma Quality Improvement Program.

be met by broadening the registry scope to be eligible for more grants.[17] This was in response to a common barrier around limited resources.[17 18 21 23 29 36 39 40 42] A trauma registry custodian explained that staff must 'make do with the resources that you have because the funding will not be there'[23] (Hashmi *et al*, p120).

**Domain 4: characteristics of individuals**

Generating staff understanding of quality improvement is needed to encourage change.[10 20 21] Increased quality improvement dissemination can help develop greater public awareness.[10 20] A barrier to this effort is the lack of staff belief in quality improvement interventions.[10 16 17 30]

A participant from a healthcare provider interview on implementing the WHO Trauma Care Checklist highlighted that 'nobody is really trying to do it as a career' as the majority 'would rather be doing joint replacements, rather than worrying about the poor people falling off trains and motorcycles'[30] (Wild, p19).

Communication training was a facilitator for TQIP implementation.[16 24 25 31 32 34] A trauma training clinician survey emphasised developing interpersonal staff communication alongside strengthening emergency communication methods.[32] Limited staff knowledge acts as a hurdle to improved communication.[10 26 31 34] A participant from a US injury prevention programme highlighted the l ack of healthy understanding of quality improvement requirements as they are viewed as 'the black sheep' of the department, making it a constant struggle to 'seek out other mentors of people that do similar work'[26] (Newcomb, p339).

### Domain 5: process of implementation

Identifying specific individuals in each organisation to drive the quality improvement intervention as a champion is significant to improve prioritisation.[17 26 30 36] Trauma registry questionnaire respondents felt that limited stakeholder engagement could be overcome by having a champion to drive and mobilise buy-in.[17] Identifying a champion is made difficult by the barrier of overworked staff.[10 16–18 31 34 37] A Tanzanian hospital was so understaffed that 'you will find you have one doctor at night and ten patients show up at once', making it difficult to document records and motivate staff[37] (Sawe, *et al*, p26).

Strengthening data quality was highlighted as a facilitator.[16 19–21 39–42] Asia-Pacific trauma leaders proposed establishing standardised minimum data requirements to collect more complete patient information and integrate injury surveillance with data collection.[19] These efforts were hindered by poor data quality across country income levels.[17–19 21 23 32 41 42] An Argentinian registry implementation project found a large variation in trauma admission protocols and widespread under-reporting, impacting the quality of clinician records.[32]

Common facilitators and barriers were arranged by income level and are summarised in table 3.

### DISCUSSION

In-hospital TQIP facilitators and barriers were similar across all income levels. Facilitators and barriers, grouped under the five CFIR domains, identified prioritising staff education and training, strengthening dialogue with stakeholders and increasing standardised guidelines for best practice as key facilitators going forward. Major barriers were the need to prioritise data quality improvement in LMICs and improved communication training in HICs. Studies focusing on in-hospital TQIP research in LMICs were limited to a few countries, indicating that the geographical scope of quality improvement research must be widened in these regions.

**Table 3** Main facilitators and barriers to TQIP implementation, by income level

| Income level | Facilitators | Barriers |
|---|---|---|
| L | ► Staff education and training<br>► Open dialogue with key stakeholders<br>► Strengthening data quality | ► Poor data quality<br>► Limited resources<br>► Overworked staff |
| LM | ► Staff education and training<br>► Tailoring to local context<br>► Strengthening data quality | ► Poor data quality<br>► Lack of buy-in<br>► Lack of common guidelines |
| UM | ► Staff education and training<br>► Tailoring to local context<br>► Prioritising buy-in | ► Poor data quality<br>► Lack of common guidelines<br>► Lack of staff knowledge |
| H | ► Staff education and training<br>► Communication training<br>► Prioritising patient care | ► Professional hierarchies<br>► Lack of common guidelines<br>► Overworked staff |

H, high; L, low; LM, lower middle; TQIP, Trauma Quality Improvement Program; UM, upper middle.

A facilitator was strengthening stakeholder dialogue to increase buy-in for in-hospital TQIPs. This supports a previous qualitative interview study that proposed adopting a hybrid quality improvement model where experts would set basic guidelines for programmes, incorporating feedback from frontline staff members according to local conditions.[43] Key barriers to stakeholder dialogue included a limited desire for TQIPs, resistance to change and competing group interests. Studies reported the need to tailor interventions to the local context, along with focusing on simplicity to improve communication. An earlier grounded theory analysis of HIC interviews identified important strategies to improve stakeholder engagement as clarity of purpose, inviting participation and engaging clinicians with feedback.[44] Given the wide range of competing innovations in hospital settings, focusing on increasing buy-in through stakeholder dialogue could strengthen the likelihood of effective implementation along with positive patient outcomes.

A reported barrier across all country income levels was the lack of common guidelines throughout the trauma care timeline. A review of LMIC trauma registry implementation highlighted that the WHO's Injury Surveillance Guidelines are only focused on collecting data on injury events rather than also addressing hospital care or

trauma outcomes.[45] The uncoordinated and resource-intensive nature of data collection acted as a barrier that contributed to staff time constraints. A survey revealed that there is no single resource available that provides a comprehensive insight into experiences across trauma registries.[46] To overcome this, quality improvement guideline standardisation was emphasised as a facilitator to improve the success of in-hospital implementation.

Limited in-hospital TQIP durability data underline the need to prioritise research on sustainable interventions. An LMIC scoping review on trauma registry implementation highlighted the need to prioritise sustainable registry development as a significant step forward.[45] Key barriers reported were the lack of long-term funding and limited buy-in for quality improvement initiatives. In our review, financial resources were not reported as widely as initially expected, particularly in LMICs. Despite this, key identified barriers such as staff time constraints and limited resources are directly linked to a lack of funding. Increased dissemination of results is a critical facilitator to establish prolonged TQIP support, through enhancing engagement and quality improvement understanding. A trauma care provider survey in the Andean region concluded that increased quality improvement dissemination can empower local providers to participate in health systems and drive change.[37] The increase in peer-reviewed quality improvement reports from LMICs emphasises dissemination through publications as a viable means of providing guidance for future programmes and expanding quality improvement knowledge.[47]

Education and training are essential facilitators to improve the chances of successful implementation. This was mentioned at all country income levels, specifically in terms of interpersonal communication in HICs. Enhancing communication channels between trauma teams to improve health outcomes is increasingly common in HICs. A quantitative study found that well-defined interdisciplinary communication between the trauma service and surgical specialists reduced the time to operation for facial trauma patients.[48] Common barriers were limited staff belief and knowledge in quality improvement interventions. While elements of the WHO's guidelines for TQIPs have been mandated by governments across income levels,[49] there have not been clear protocols put into place for implementing, evaluating and sustaining interventions. This makes it difficult to provide standardised training for staff, which can impact the quality of patient care.

Enhancing data quality is a crucial facilitator to strengthening in-hospital implementation. Trauma registries were the most common in-hospital TQIP intervention, particularly in LMICs, with studies citing poor data quality as the main barrier. Reasons for the lack of data quality centred on high rates of missing data and limited standardised variables for comparison. A scoping review assessing trauma registry data quality found that only 4 out of 69 publications provided a general classification of data quality, in which the taxonomy was inconsistent.[50]

A similar systematic review identified only 10 studies that evaluated data quality in trauma registries.[51] The majority assessed data quality based on completeness, with large differences reported between papers, and no evidence found on data precision or timeliness.[51] This lack of uniformity makes it difficult to draw insights on the quality of trauma care delivered and compare across country income levels. The Lancet Global Health Commission on High Quality Health Systems underlined the disparity between the growing global injury burden and the limited data availability on care quality.[52] Providing incentives for data entry participation, such as hospital-specific dashboards or research involvement, could improve data quality and support effective in-hospital implementation.

## Strengths and limitations

The review covered a broad range of in-hospital TQIP interventions and included studies with any design. The screening process was conducted along with two peer reviewers, which allowed for comparison and greater input in decision-making—increasing protection against potential bias. This review provides a snapshot of the research field through a methodologically robust process, which includes a transparent method of reviewing in a limited time frame.[53] The review captured a large number of studies for analysis from various databases.

The use of three databases and the restriction of literature to English-language studies could have biased the findings if excluded studies were significantly different to those included in the scoping review. The use of more databases, including preprint servers, a wider time frame and the inclusion of a greater number of studies could have provided data that may have been neglected. We attempted to limit the selection bias by consulting a librarian with extensive experience conducting scoping reviews when selecting databases.

Grouping literature into income levels by country may provide another limitation. This classification does not capture the complexity of resource allocation within countries, reducing the generalisability of the results. The scoping review process does not place as much emphasis on assessing the quality of included studies when compared with systematic reviews.[54] The lack of quality assessment limits the implications of gathered data as the study focus is primarily concerned with mapping available literature on the topic. There is a risk of researcher bias as the process of choosing and analysing literature can be subjective. Important steps to avoid researcher bias were focusing on the research question, following the eligibility criteria and cross-referencing screening selection with a peer reviewer.

## CONCLUSION

Staff education and training, strengthening stakeholder dialogue and increasing standardised trauma care guidelines were reported as the main facilitators, while major barriers identified were poor data quality in LMICs and

the lack of effective communication training in HICs. Stakeholder prioritisation of TQIPs, along with increased knowledge and consensus on trauma care best practice, could further advance efforts to lower the global trauma burden. The focus of future in-hospital TQIPs in LMICs should primarily be concerned with improving the data quality of registries, while interventions in HICs should focus on the communication skills of healthcare professionals.

**Acknowledgements** We would like to acknowledge Karolinska Institutet Library (Independent Information Specialist Consultants, Solna, Sweden) for peer review of the literature search strategy, along with the faculty and students of Karolinska Institutet Department of Global Public Health for comments and feedback on earlier versions of this manuscript.

**Contributors** Guarantor for overall content of study - GK. Study concept, design and search strategy—GK, JB and MGW. Data selection and screening—GK, JB, YS and MGW. Data analysis and interpretation—GK, JB and MGW. Drafting of the manuscript—GK, JB and MGW. Critical revisions of the manuscript—GK, JB, YS and MGW. Study supervision—JB and MGW. All authors read and approved the final manuscript.

**Funding** The authors have not declared a specific grant for this research from any funding agency in the public, commercial or not-for-profit sectors.

**Competing interests** None declared.

**Patient and public involvement** Patients and/or the public were not involved in the design, or conduct, or reporting, or dissemination plans of this research.

**Patient consent for publication** Not required.

**Ethics approval** Not applicable.

**Provenance and peer review** Not commissioned; externally peer reviewed.

**Data availability statement** All data relevant to the study are included in the article or uploaded as supplemental information. The data that support the findings of this study are available from PubMed, Web of Science and Global Index Medicus databases using the search string outlined in online supplemental file 1.

**ORCID iDs**
George Kapanadze http://orcid.org/0000-0002-3199-1252
Johanna Berg http://orcid.org/0000-0001-7553-7337
Martin Gerdin Wärnberg http://orcid.org/0000-0001-6069-4794

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
