## [Reviewer comments · BMJ Open]

ARTICLE DETAILS

TITLE (PROVISIONAL)	Facilitators and barriers impacting in-hospital Trauma Quality Improvement Program (TQIP) implementation across country income levels: a scoping review
AUTHORS	Kapanadze, George; Berg, Johanna; Sun, Yue; Gerdin Wärmberg, Martin

VERSION 1 – REVIEW

REVIEWER	Barnett, Adrian Queensland University of Technology, Institute of Health and Biomedical Innovation
REVIEW RETURNED	08-Nov-2022

GENERAL COMMENTS	This scoping review examined trauma and the use of quality improvement programs across a range of countries with varying average income levels. It used PRISMA to help complete the stages of the review and CFIR to provide structure on the implementation characteristics. It was well structured, well written, and appears to be well conducted. The results indicated a lack of support for this "unglamorous" activity of quality improvement, which was interesting and not surprising. I was surprised that there wasn't anything about the evidence behind the interventions. "Evidence strength and quality" is in the CFIR. Was this a very low priority in this context? Some of the barriers discussed about other priorities and lack of resources seemed like reasonable reasons to refuse doing any extra work on, for example, a registry. Hospitals will have their own cost-benefit trade-offs, and there could be competing innovations that would give a better outcome for patients. Data quality was a repeated issue from the review, and this is not just in LMICs, as hospital data from high income countries often include missingness and errors. Getting something back for entering the data might help with encouraging participation, such as hospital-specific dashboards or the potential to be involved in research. Minor comments - The search criteria were clear and looked appropriate.- There are a lot of acronyms in the paper. I don't think quality improvement needs to be an acronym.- Would it be useful to have an example of a specific TQIP in the introduction? I jumped ahead to look at one of the included papers so that I had an idea about the kinds of interventions.
---

	 - Line 42, "unpublished" means not in a journal? There could be preprints that are good quality, although they may not have been peer reviewed. - "Most of the studies originated in upper middle income countries (7 [25%])" the percentages here for the four country categories do not add to 100%, perhaps because some studies crossed income categories? - These acronyms in table 1 are not explained: IP, IPC, M&M
--	--

REVIEWER	Herrera-Escobar, Juan Brigham and Women's Hospital Department of Surgery, Center for Surgery and Public Health
REVIEW RETURNED	19-Dec-2022

GENERAL COMMENTS	This scoping review of the literature aims to identify key facilitators and barriers to trauma quality improvement programs (TQIP) implementation across countries' income levels. The authors identified 28 studies and found that the key facilitators and barriers were the need to prioritize staff education and training, strengthen dialogue with stakeholders, and provide standardized best-practice guidelines. This is a nice review on an interesting topic; however, there are a few things I would like the authors to consider. Comments:  1. The title is lengthy and redundant. I suggest: Facilitators and barriers impacting in-hospital TQIP implementation: A scoping review. 2. It is confusing when the authors refer to "across income levels". Income levels of what? Countries? Regions? Systems? Hospitals? Authors should be aware that resources vary significantly within each of these. As such, the unit of analysis should be clearly specified in the objective. 3. Is the need to prioritize staff education and training a barrier? Is the need to strengthen dialogue with stakeholders a barrier? I think these are the gaps, not the barriers. A barrier would be something preventing those things from happening. Authors should review the terminology they used in order to get their message across. 4. What does "Data quality improvements were more apparent in LMICs" mean? Also, the abstract could be significantly improved to better reflect the findings of the manuscript. 5. I'm surprised that the financial aspect of maintaining TQIPs is only briefly mentioned/discussed a couple of times in the manuscript, in particular for LMICs. Is this because the included studies did not mention it much? If so, the reasons for this should be discussed.
--

VERSION 1 – AUTHOR RESPONSE

Comments from Reviewer #1

1. Overall reviewer comments: This scoping review examined trauma and the use of quality improvement programs across a range of countries with varying average income levels. It used PRISMA to help complete the stages of the review and CFIR to provide structure on the implementation characteristics. It was well structured, well written, and appears to be well conducted. The results indicated a lack of support for this "unglamorous" activity of quality improvement, which was interesting and not surprising.

Response: We highly appreciate the positive feedback and value the in-depth assessment along with associated suggestions.

2. I was surprised that there wasn't anything about the evidence behind the interventions. "Evidence strength and quality" is in the CFIR. Was this a very low priority in this context?

Response: Thank you for the insightful comment. We agree that the evidence behind the interventions is important, especially when deciding whether to implement a specific TQIP, but beyond the scope of this review. We focused on the facilitators and barriers of implementing TQIP, once it had been decided that a specific program would be implemented. We have added a sentence at the end of the data charting subsection in the methods clarifying that "we did not evaluate the evidence underlying the identified TQIPs, as our focus was on facilitators and barriers to implementing a TQIP once attempted."

3. Some of the barriers discussed about other priorities and lack of resources seemed like reasonable reasons to refuse doing any extra work on, for example, a registry. Hospitals will have their own cost-benefit trade-offs, and there could be competing innovations that would give a better outcome for patients.

Response: We appreciate your insights on this. We have added a sentence to the end of the strengthening stakeholder dialogue paragraph in the discussion on pages 18-19 given your analysis. The additional sentence reads "Given the wide range of competing innovations in hospital settings, focusing on increasing buy-in through stakeholder dialogue could strengthen the likelihood of effective implementation along with positive patient outcomes."

4. Data quality was a repeated issue from the review, and this is not just in LMICs, as hospital data from high income countries often include missingness and errors. Getting something back for entering the data might help with encouraging participation, such as hospital-specific dashboards or the potential to be involved in research.

Response: Thank you for your input on this. We have included a sentence at the end of the enhancing data quality paragraph in the discussion on page 20 to supplement your feedback. The sentence reads "Providing incentives for data entry participation, such as hospital-specific dashboards or research involvement, could improve data quality and support effective in-hospital implementation."

5. The search criteria were clear and looked appropriate.

Response: We appreciate your positive feedback.

6. There are a lot of acronyms in the paper. I don't think quality improvement needs to be an acronym.

Response: Thank you for pointing this out. We have removed quality improvement as an abbreviation from the table on page 22 and have changed all subsequent QI abbreviations in the article back to quality improvement.

7. Would it be useful to have an example of a specific TQIP in the introduction? I jumped ahead to look at one of the included papers so that I had an idea about the kinds of interventions.

Response: We appreciate this observation. To provide a concrete example of some in-hospital TQIPs we will discuss in the article, a sentence has been added in the first paragraph of the introduction on page 5. The sentence reads “Key in-hospital TQIPs highlighted by the World Health Organisation (WHO) in their 2009 guidelines included techniques such as trauma registries along with mortality and morbidity conferences.”

8. Line 42, "unpublished" means not in a journal? There could be preprints that are good quality, although they may not have been peer reviewed.

Response: Thank you for the question, indeed “unpublished” refers to literature not in a journal. Given the time and scope restraints of the review, it was decided that the focus would be placed on peer-reviewed articles. Since we did not assess the quality of each article, we felt that this was the most appropriate action to take, but we added “including preprint servers” to the sentence on conducting a wider search in the limitations section on page 21.

9. "Most of the studies originated in upper middle income countries (7 [25%])" the percentages here for the four country categories do not add to 100%, perhaps because some studies crossed income categories?

Response: We appreciate your insight on this. We have added a sentence in the first paragraph of the results section on page 9 to clarify that a significant number of the studies included in the review crossed multiple income levels. The sentence reads “The remaining studies (8 [29%]) included multi-country analyses that spanned across income levels.”

10. These acronyms in table 1 are not explained: IP, IPC, M&M.

Response: Thank you for mentioning this. Table 1 on pages 9-11 has been edited to remove the three abbreviations mentioned above and replace them with the full form names - “Injury Prevention”, “Injury Prevention Champion”, and “Mortality & Morbidity”.

Comments from Reviewer #2

1. Overall reviewer comments: This scoping review of the literature aims to identify key facilitators and barriers to trauma quality improvement programs (TQIP) implementation across countries' income levels. The authors identified 28 studies and found that the key facilitators and barriers were the need to prioritize staff education and training, strengthen dialogue with stakeholders, and provide standardized best-practice guidelines. This is a nice review on an interesting topic; however, there are a few things I would like the authors to consider.

Response: We are grateful for your detailed and insightful feedback along with your relevant suggestions.

2. The title is lengthy and redundant. I suggest: Facilitators and barriers impacting in-hospital TQIP implementation: A scoping review.

Response: Thank you for your suggestion. The article title has been changed to “Facilitators and barriers impacting in-hospital Trauma Quality Improvement Program (TQIP) implementation across country income levels: A scoping review.”

3. It is confusing when the authors refer to “across income levels”. Income levels of what? Countries? Regions? Systems? Hospitals? Authors should be aware that resources vary significantly within each of these. As such, the unit of analysis should be clearly specified in the objective.

Response: We appreciate your important feedback on this. We have clarified that our unit of analysis is at the country level by adding “country income level” throughout the article where appropriate along with adjusting the title accordingly. In addition, we have added a methodological limitation point in the ‘Strengths and limitations of this study’ section on page 4 that reads “The generalisability of our results was impacted by grouping literature based on country income level at the time of publication as we did not investigate potential differences in available resources within countries (e.g., regions and healthcare institutions).”

4. Is the need to prioritize staff education and training a barrier? Is the need to strengthen dialogue with stakeholders a barrier? I think these are the gaps, not the barriers. A barrier would be something preventing those things from happening. Authors should review the terminology they used in order to get their message across.

Response: Thank you for the suggestions. To clarify, the above-mentioned factors are not listed as barriers but instead as facilitators. Table 2 on pages 11-13 highlights key barriers that influence the success of facilitators - e.g. “lack of buy-in” from leadership for stakeholder engagement and a “limited desire for quality improvement initiatives” for prioritising staff education and training.

5. What does “Data quality improvements were more apparent in LMICs” mean?

Response: We appreciate your input on this. We have changed the wording in the results section of the abstract on pages 2-3 to read “the key barrier theme identified in LMICs was poor data quality.” By this, we mean that “poor data quality” was the barrier theme that came up most often from the studies originating from countries in the LMIC setting. This does not exclude that “poor data quality” is also an issue in high income countries, but this was not as prevalent in the reviewed literature compared to LMICs.

6. The abstract could be significantly improved to better reflect the findings of the manuscript.

Response: Thank you for the insight. The abstract has been re-formatted and lengthened on pages 2-3 to more closely match the findings of the article.

7. I’m surprised that the financial aspect of maintaining TQIPs is only briefly mentioned/discussed a couple of times in the manuscript, in particular for LMICs. Is this because the included studies did not mention it much? If so, the reasons for this should be discussed.

Response: We are grateful for this observation. We found that the financial component of maintaining in-hospital TQIPs was not mentioned as frequently as initially expected prior to the review. Instead, there was a focus on general “resources” that also included human factors such as a lack of “specialised personnel” in LMICs - as seen in table 2 on page 13. We have added a couple of sentences in the discussion on page 19 that underlines the fundamental impact of finances on maintaining TQIPs. The addition reads as “In our review, financial resources were not reported as widely as initially expected, particularly in LMICs. Despite this, key identified barriers such as staff time constraints and limited resources are directly linked to a lack of funding.”

VERSION 2 – REVIEW

REVIEWER	Barnett, Adrian Queensland University of Technology, Institute of Health and Biomedical Innovation
REVIEW RETURNED	24-Jan-2023

GENERAL COMMENTS	I was surprised that evidence was not included, especially as other intervention characteristics were included. I note the authors answer about evaluating already chosen TQIFs. Minor comments  - I agree about the importance of simplicity. I have seen similar data collection processes fail because of too many fields that are too complex. - “Subject” on page 14 should be “participant” - Sentence needs re-wording, something missing? “Identifying staff to act as a champion for the quality improvement intervention is significant.” - For the last sentence on page 15, it’s not clear if it’s asking for more research in LMICs or more quality improvement activities. As the study examined research, then the statement should probably only cover research. - Can "decreased mortality outcomes" simply be 'decreased deaths'
---

REVIEWER	Herrera-Escobar, Juan Brigham and Women's Hospital Department of Surgery, Center for Surgery and Public Health
REVIEW RETURNED	29-Jan-2023

GENERAL COMMENTS	The authors have satisfactorily addressed all my comments
---

VERSION 2 – AUTHOR RESPONSE

Comments from Reviewer #1

1. I was surprised that evidence was not included, especially as other intervention characteristics were included. I note the authors' answer about evaluating already chosen TQIPs.

Response: We appreciate this comment. As the aim of our manuscript was to identify facilitators and barriers to TQIP implementation we felt that it was most appropriate to focus on addressing established initiatives. We take on board your feedback and believe that evaluating evidence for TQIPs is an important area that requires further attention in future studies.

2. I agree about the importance of simplicity. I have seen similar data collection processes fail because of too many fields that are too complex.

Response: Thank you for the positive response. Increased simplicity of data collection was a significant facilitator mentioned in numerous studies, we agree that this is a vital component of effective quality improvement programs.

3. “Subject” on page 14 should be “participant”

Response: We appreciate you pointing this out. The wording has been changed to “participant” on page 14.

4. The sentence needs re-wording, is something missing? "Identifying staff to act as a champion for the quality improvement intervention is significant."

Response: Thank you for drawing attention to this. The sentence was included to highlight the need for designated individuals in each organisation to take up the task of driving quality improvement initiatives to improve prioritisation. To clarify this, we have changed the sentence to "Identifying specific individuals in each organisation to drive the quality improvement intervention as a champion is significant to improve prioritisation" on page 14.

5. For the last sentence on page 15, it's not clear if it's asking for more research in LMICs or more quality improvement activities. As the study examined research, then the statement should probably only cover research.

Response: We appreciate the feedback on this point. The sentence was geared more toward the research component in LMICs but we understand that this may have been unclear. Accordingly, we have changed the sentence to "studies focusing on in-hospital TQIP research in LMICs were limited to a few countries, indicating that the geographic scope of quality improvement research must be widened in these regions" on page 15.

6. Can "decreased mortality outcomes" simply be 'decreased deaths'.

Response: Thank you for this insightful comment. The sentence regarding 'decreased mortality outcomes' was removed from the manuscript during the first revision to improve the clarity of the paper.

Comments from Reviewer #2

1. The authors have satisfactorily addressed all my comments.

Response: We are grateful for your positive feedback and appreciate the time and effort you have put into reviewing our manuscript.